# Host Dynamics under General-Purpose Force Fields

**DOI:** 10.3390/molecules28165940

**Published:** 2023-08-08

**Authors:** Xiaohui Wang, Zhe Huai, Zhaoxi Sun

**Affiliations:** 1Beijing Leto Laboratories Co., Ltd., Beijing 100083, China; 2College of Chemistry and Molecular Engineering, Peking University, Beijing 100871, China; 3XtalPi—AI Research Center, 7F, Tower A, Dongsheng Building, No. 8, Zhongguancun East Road, Beijing 100083, China

**Keywords:** macrocyclic hosts, force field development, ab initio calculations, conformational preference, atomic charges

## Abstract

Macrocyclic hosts as prototypical receptors to gaseous and drug-like guests are crucial components in pharmaceutical research. The external guests are often coordinated at the center of these macromolecular containers. The formation of host–guest coordination is accompanied by the broken of host–water and host–ion interactions and sometimes also involves some conformational rearrangements of the host. A balanced description of various components of interacting terms is indispensable. However, up to now, the modeling community still lacks a general yet detailed understanding of commonly employed general-purpose force fields and the host dynamics produced by these popular selections. To fill this critical gap, in this paper, we profile the energetics and dynamics of four types of popular macrocycles, including cucurbiturils, pillararenes, cyclodextrins, and octa acids. The presented investigations of force field definitions, refitting, and evaluations are unprecedently detailed. Based on the valuable observations and insightful explanations, we finally summarize some general guidelines on force field parametrization and selection in host–guest modeling.

## 1. Introduction

Macrocyclic hosts are employed as drug carriers and reservoirs in pharmaceutical applications [1,2]. The external drug-like molecules, historically recognized as guests to the host cavity, are often coordinated at the center of the host molecule. The formation of host–guest interactions could be used to protect drugs that are unstable under specific conditions (e.g., light-induced damage) or improve the physiochemical properties (e.g., solubility) of pharmaceutically active molecules [3,4,5,6]. Although these macromolecular containers are commonly considered prototypical receptors compared with biomacromolecules, their dynamic behaviors and interactions with external guests could be ‘counterintuitively’ complex, featuring multi-modal binding behaviors due to the interplay between intra-molecular conformational preferences of individual components and the inter-molecular directional coordination between functional groups [7,8,9,10]. Due to the time scale of inter-molecular packing involving these macrocyclic hosts, fixed-charge force fields are the mainstream description in molecular modeling [11,12,13,14]. Common choices for electrostatics are AM1-BCC [15] and RESP [16] charge schemes, which parametrize atomic charges via molecule-specific geometric optimization, electrostatic potential (ESP) scan, and regularized least-squares fitting. By contrast, the other parameters, due to their relatively small system dependence and the relatively high computational burden in fitting these terms, refs. [17,18,19,20] are always grabbed from pre-fitted transferable parameter sets such as general AMBER force field (GAFF) [21] derivatives. Despite the massive application of these popular practices, there is seldom a detailed perspective of these general-purpose force fields that provides critical insights into their accuracy levels and the dynamic behaviors produced by them. Direct applications without justification could be rather dangerous, and the simulation outcome could be untrustworthy. Although in many scientific reports, the simulation outcomes of some specific properties, such as binding thermodynamics, coincidentally coincide with experimental reference, these lucky shots could be poisonous to this field due to the acceptance of fortuitous and possibly erroneous error cancellations. To avoid the disturbance of an unvalidated simulation foundation and provide practically useful insights into the selection and parametrization of fixed-charge force fields, we devote the current paper to an unprecedently detailed investigation of common practices in the host–guest modeling community, featuring charge-quality assessment, force field definition comparison, molecule-specific bonded-parameter refitting, energetic and force errors benchmarking, and host dynamics probing. The accumulated detailed yet general picture of force field behaviors secures valuable guidelines on force field parametrization in host–guest modeling.

Four popular host families are considered in this work. Cucurbit[*n*]urils (CB[*n*] or CB*n*), pillar[*n*]arenes, and cyclodextrins (CD) are symmetrical two-rim macrocycles with different compositions (e.g., substitutions and heteroatoms), while the other host family octa acid (OA) features a basket-like single-entrance pocket. For macrocyclic hosts, the number of repeating units as a tunable parameter is used to tune the cavity volume and entrance size. As the prototypical hosts often exhibit non-satisfactory physiochemical behaviors, chemical modifications are often imposed on the host structure in order to fine-tune the host properties, e.g., solubility, guest-binding strength, and entrance size. To cover the commonly employed species from the four host families, we carefully select molecules with representative features as investigation targets, which are summarized in Figure 1. For example, the 6-unit, 7-unit, and 8-unit cucurbiturils cover commonly applied species and yet involve a variation of the cavity size, the prototypical β-CD and its methylated form Heptakis-2,6-DiMe-β-CD (Hβ-CD) differ in both the hydrogen-bonding ability and the entrance size, and the prototype OA and its methylated form tetra-endo-methyl octa acid (TEMOA) involve a variation of the entrance size.

## 2. Electrostatics

Atomistic fixed-charge modeling models electrostatics with a charge–charge Coulomb interaction. To accurately reproduce ab initio results, often the accurate reproduction of ESP around each molecule is pursued. Commonly employed charge schemes include AM1-BCC [15] and RESP [16]. The former regime approximates the HF/6-31G* ESP by combining gas-phase AM1 Mulliken charges with a knowledge-based term derived from connectivity information (i.e., BCC), while the latter directly performs ab initio calculations, scans the molecular ESP (traditionally at HF/6-31G*), and optimizes a least-squares loss with hyperbolic regularizations to improve the robustness and decrease the sensitivity to conformational variations. Concerning computational efficiency, AM1-BCC is much faster than RESP. However, as the BCC correction term is fitted with drug-like molecules, significant errors could be introduced if the studied molecules show noticeable differences from the training set. This fact has been recognized in 8-unit cucurbiturils and 6-unit cyclodextrins in our previous works, refs. [7,8] and in this paper, we will investigate whether the conclusion could be generalized to most macrocycles.

Due to the structural similarities of hosts from the same family, we pick a subset of host molecules in Figure 1 according to representative structural features and form the charge-evaluation host set. Specifically, we consider CB7 and CB8 from the cucurbituril family due to their different ring sizes, only SP6 from the pillararene family due to the similarities of SP*n* in Figure 1 and the existing ring-size dependent investigation in cucurbiturils, OA and TEMOA from the OA family due to their different entrance sizes, and β-CD and Hβ-CD from cyclodextrins due to the substituting methyl groups.

As discussed above, the ultimate goal in charge fitting is to reproduce molecular ESP rather than any specific charge distribution. Thus, the quality of ESP reproduction serves as a crucial factor in the evaluation of charge quality. The charge-generated molecular ESP is compared with the fitting target (i.e., HF/6-31G* ESP), and the error is quantified by a commonly used metric named relative root-mean-squared error (RRMSE). Aside from the traditional HF/6-31G* level for ESP scan, we also consider many other modern options, including B3LYP [22,23,24], BP86 [25,26], PBE [27], PW6B95 [28], TPSSh [29,30], and wB97X-D [31], all of which are combined with a rather upgraded huge basis set def2-QZVPP [32,33], due to many criticisms on the use of the ‘crude’ traditional HF/6-31G*. The resulting def2-QZVPP-combined ab initio levels are very costly even with modern computing resources, especially for huge molecules like macrocyclic hosts considered in the current work. Therefore, the high-level huge-basis-set results serve as a practically converged fitting target in host parametrization and benchmarking reference in quality evaluation.

The generated charge sets include AM1-BCC and two RESP charge sets. The first RESP charge set is fitted with the traditional HF/6-31G* data, while the second one is obtained with another ab initio reference PW6B96/def2-QZVPP. All of the three charge sets are then used to compute the ESP around each molecule, with which the ab initio ESP are compared. The numerical ESP RRMSE data are presented in Figure 2. We first present a comparison across the charge schemes and then investigate the system-specific behaviors in detail. For AM1-BCC, the ESP RRMSE values are obviously larger than the other two RESP charge sets. In Figure 2a, the ESP deviations from the HF/6-31G* data are smaller compared with those from the other higher-level def2-QZVPP references, which agrees with the fact that the semi-empirical charge scheme is trained with a database of HF/6-31G* ESP of drug-like molecules. The AM1-BCC regime, interestingly, does not incur significant problems for the OA family and SP6, which indicates the applicability of this semi-empirical charge regime in these molecules. However, considering the considerable problems in cucurbiturils and cyclodextrins, it is preferable to shift to the more rigorous RESP fitting in general for host–guest modeling.

When the RESP charge scheme is employed, the magnitude of ESP RRMSE decreases, although some charge-set-dependent behaviors are still observed (c.f., Figure 2b,c). For the OA family and the pillararene family, the ESP deviations are quite small for both RESP charge sets, which indicates the absence of numerical problems in the regularized fitting. This phenomenon is not unexpected, as these molecules do not have sites/groups that generate significantly anisotropic ESP distribution. By contrast, both the cucurbituril family and the cyclodextrin family exhibit significant ESP errors, but the reasons causing these deviations differ. For cucurbiturils, if we only consider the fitting target as the reference in evaluation (i.e., HF/6-31G* columns in Figure 2b and PW6B95/def2-QZVPP columns in Figure 2c), the ESP RRMSE values are satisfactorily small (~5%). However, when the evaluation reference differs from the fitting target, the ESP deviations increase. Specifically, for the HF/6-31G*-targeted RESP charges, the ESP deviations from DFT references are increased to ~18%, while the increases for the PW6B65/def2-QZVPP-targeted RESP charges are smaller. Based on this observation, we know that the DFT-generated ESP data are similar but differ noticeably from the HF/6-31G* ESP. It is difficult to conclude which ab initio reference is a better choice with only the ESP data, as these results are generated for a standalone molecule in vacuo, while in practical situations, the molecule could form complex interactions with external agents and surrounding solvent molecules, which could involve many other influential factors, e.g., polarization and charge transfer.

The huge ESP reproduction problem in cyclodextrins, by contrast, involves further factors. As we can see in Figure 2b,c, the ESP RRMSE values for cyclodextrins are larger than cucurbiturils. The level of theory for the ESP scan plays a role here, but another critical problem is the numerical difficulty in the least-squares fitting. Specifically, the ESP distribution close to the large number of -OH groups in cyclodextrins cannot be accurately reproduced with atom-centered fixed-charge models. The RESP procedure optimizes the values of atomic charges to reproduce ab initio ESP in the presence of hyperbolic regularizations to avoid overfitting and minimize conformational dependence. However, atom-centered charges cannot accurately reproduce anisotropic ESP distributions unless virtual sites for lone pairs or other treatments are incorporated. Therefore, to accurately describe the electrostatics involving cyclodextrin derivatives, more detailed descriptions are required.

Overall, for the accurate description of host electrostatics with fixed-charge models, the RESP charge scheme prevails and should be employed. Two key factors in RESP fitting are the selection of ab initio level for ESP scan and the anisotropic ESP distribution. For the former point, HF/6-31G* and higher-level selections could be applied, but in many cases, the traditional HF/6-31G* level already seems acceptable, and there is no solid evidence supporting the usage of any higher-level option. As for the latter point, the presence of such numerical problems could be identified based on the structure of the host. Commonly encountered centers/groups that trigger anisotropic ESP distributions are -OH and -NH_2_. If these groups are in their neutral forms, numerical problems would be triggered in charge fitting. Upon the deprotonation of -OH or the protonation of -NH_2_, the anisotropic behavior vanishes, and the ESP distribution could be reproduced with atom-centered models. Finally, it would be beneficial to note that accurate ESP reproduction, although being rigorous in the intended use of the RESP charge scheme, does not guarantee the accurate description of inter-molecular electrostatics due to the involvement of many other influencing factors, e.g., charge transfer and polarization effects. A popular example is ionic solvents, where numerous highly charged species are packed together. In these situations, additional alterations are required to accurately consider these remaining influential effects [34,35,36].

## 3. Bonded and vdW Terms

After detailing the differences and similarities of atomic charges generated with different procedures, we then turn to bonded and vdW terms in general-purpose force fields. The comparison in this section consists of three sub-stages. The first stage features a term-specific comparison between the two popular GAFF versions, which deepens our understanding of commonly employed parameter sets and their ‘preferred’ conformational space. Also, such a term-wise comparison provides useful hints on possible differences in host dynamics. Next, the second stage involves hundreds of ns explicit-solvent sampling for each host molecule, which provides direct observations of the collective effects of all force field terms. In the last stage, the general-purpose force fields are refitted in a molecule-specific manner with the generalized force-matching (FM) [37,38] scheme in order to reach the accuracy limit under the given functional form of the force field. The deviations of energetics and atomic force from an accurate and robust ab initio reference are computed as an evaluation of the closeness of general-purpose force fields to the high-level reference, and dynamics simulations are then conducted to probe the impact of parameter adjustment on host dynamics, presenting the real dynamic behavior of macrocycles. Below, we first detail the force field refitting and evaluation procedure, as the same workflow is applied to all host families studied in this work.

We first briefly outline the refitting procedure. The transferable GAFF(2) parameters of the host molecule are optimized in a molecule-specific manner in order to achieve higher accuracy. The least-squares loss function includes the system energy, atomic force (Frobenius norm), and a weak harmonic (L2) regularization term restraining the parameter space not too far from the initial guess to avoid overfitting. As this regularization term is initial-guess dependent, the whole force field refitting procedure is repeated with the two GAFF versions. The non-bonded terms, 1–4 scaling factors, the dihedral periodicity, and the atom-type definitions remain untouched, and only bonded terms (i.e., bond stretching, angle bending, and torsional terms) are under adjustment. The reference data is accumulated with gas-phase sampling and ab initio calculations in an adaptive fashion. Specifically, we accumulate 500 configurations per iteration and optimize the parameter set with existing data. The sampling is performed at 600 K to enable an unconstrained exploration of the configurational space, and the sampling interval is 10 ps. In total, 18 iterations (5 ns each) are performed, and the training set contains 9000 configurations. The reference ab initio data are generated with the composite method B97-3c, [39] which serves as a selection with balanced efficiency and accuracy in the calculation of various observables (e.g., geometries and thermodynamics) for practical chemical systems. All ab initio calculations are performed with the ORCA [40,41] package.

After obtaining a molecule-specific force field refitted from GAFF(2), we perform further gas-phase sampling to construct the testing set for force field evaluation. The sampling protocol is similar to the training set, but this time the sampling length is limited to 5 ns (500 configurations), and the sampling temperature is lowered to 300 K in order to only assess the low-energy room-temperature-accessible region in the conformational space. Note that the parameter set used in this stage is the final optimized FM-B97-3c force field. Based on the accumulated configurations, we compute the energy and force errors of each parameter set relative to the ab initio reference used in parameter optimization (i.e., B97-3c). Two error metrics, including the root-mean-squared error (RMSE) and the mean absolute error (MAE), are used for quality assessment. A final note is that all parameter files of macrocyclic hosts before and after refitting are provided online at https://github.com/proszxppp/host-dynamics (accessed on 1 July 2023) for interested readers to probe the details of the force field differences and parameter variations upon regularized refitting and apply these high-quality parameter sets in their own research.

### 3.1. Cucurbiturils

The first host family under investigation is CB[*n*]. This is a relatively well-studied host family with extensive experience in host–guest binding features and thermodynamics [8,42,43,44]. However, the host dynamics are not well understood, although some detailed investigations of force field definitions and evaluations have been reported for CB8 in our previous work [42]. To fully understand the host dynamics under general-purpose force fields, we choose three popular species from this family, including CB6, CB7, and CB8. Among them, CB7 and CB8 are more popular due to the compatibility of their cavity size/volume with most compounds of pharmaceutical activities.

#### 3.1.1. Term-by-Term Comparison and Host Dynamics under Transferable Parameter Sets

The first step in force field comparison is investigating the detailed definitions of force field terms. Using the CB7 host as an example, we identify two torsional terms shown in Figure 3a,b, bond-stretching and angle-bending potentials and vdW parameters that exhibit noticeable differences. As vdW potentials play a relatively insignificant role in intra-molecular interactions and the harmonic bond-stretching and angle-bending potentials influence the conformational stiffness of the host ring in an indirect manner, we focus on torsional terms in this comparison. For the torsion shown in Figure 3a, in GAFF, there is no explicit definition for this term, and the stiffness and conformational preference in this region are mainly determined by other torsional terms neighboring this region. By contrast, in GAFF2 the torsional barrier is explicitly defined as ~2.08 kcal/mol with a periodicity of 2. As a result, the GAFF2 parameter set is expected to be stiffer in this region compared with GAFF. As for the torsion in Figure 3b, although in both GAFF and GAFF2, the torsional potentials for this region are defined, the phases and barrier heights are altered, which would also impact the host dynamics in the region. The above differences have also been observed in our previous work investigating the CB8 host [42] and are general for the CB[*n*] family.

After obtaining a term-specific understanding of the similarities and differences of GAFF derivatives, we then perform 500 ns unbiased simulations in explicit water to probe the dynamics of the host molecules. For this simulation, we use GROMACS 2020.6 [45], the velocity rescaling algorithm [46] for temperature regulation at 300 K and the Parrinello–Rahman barostat [47,48] for pressure regulation at 1 atm, a time step of 1 fs, smooth particle-mesh Ewald [49] for long-range electrostatics. The host structures are superposed with 500 ps per snapshot for clarity in Figure 3c. Through the GAFF-vs-GAFF2 comparison of host dynamics, an interesting phenomenon is observed. For the 6-unit cycle CB6, the host dynamics under GAFF and GAFF2 are almost the same. The host remains in the opened-cavity state in the whole 500 ns sampling. When the number of repeating units increases to seven (i.e., CB7), there is still no noticeable difference between GAFF and GAFF2 dynamics. Interestingly, when the number of repeating units increases further to the eight-unit member CB8, the host dynamics begin to exhibit noticeable differences under the two GAFF derivatives. The GAFF host interconverts between the fully opened state and the squashed state periodically, while under GAFF2, the host remains in the open state. Similar conclusions could also be reached from the time series of the radius of gyration (Rg) shown in Figure 3d, where the GAFF and GAFF2 curves are similar for CB6 and CB7 but exhibit noticeable differences for CB8.

The relatively flexible behavior under GAFF and the stiffer GAFF2 host are expected and coincide with the analysis of torsional potentials, but the similar host dynamics under GAFF and GAFF2 for CB6 and CB7 are rather unexpected. Thus, differences in torsional terms do not necessarily lead to differences in the dynamic behaviors of macrocyclic hosts, and we cannot simply infer the host dynamics from force field definitions. From CB6 to CB8, the number of repeating units is increased, which is accompanied by an increase in the ring flexibility (or, equivalently, the decrease of the internal strain of the macrocycle). Note that the flexibility discussed here refers to the intrinsic flexibility of the host ring rather than that defined by torsional potentials. The real dynamics are determined by both the intrinsic flexibility of the host ring and the force field definitions. Following this flexibility-increasing trend, we could further infer that species in this family with even larger numbers of repeating units (e.g., CB10) could exhibit more complex dynamic behaviors under general-purpose force fields.

#### 3.1.2. Refitting and Evaluation

The above-detailed comparison of force field definitions confirms the noticeable differences between the two commonly employed GAFF derivatives for cucurbiturils, but whether the differences in torsional potentials would trigger alterations of dynamic behaviors is related to the intrinsic flexibility of the host ring. Only when the number of repeating units hits eight does the torsional potentials begin to incur noticeable deviations. For CB7, the dynamic properties under GAFF and GAFF2 are very similar, but it would be interesting to evaluate their closeness to higher-level Hamiltonians and identify problematic parameters in order to provide a direct energetic perspective of the force field quality. Also, we refit the transferable parameter set, expecting to reach the accuracy limit of the functional form of the force field and secure a high-accuracy close-to-ab initio description. The details of the force field refitting and evaluation have been provided at the beginning of Section 3. Both GAFF versions are employed as the initial guess and restraint reference, and the resulting refitted parameter sets are named FM-B97-3c-from-GAFF and FM-B97-3c-from-GAFF2.

We first present the quality of energetics produced by the general-purpose GAFF(2) and the refitted force fields in Appendix A. It is clearly shown that the pre-fitted GAFF and GAFF2 produce larger deviations from the ab initio reference than the refitted parameter sets. The large RMSE is cut in half upon regularized parameter adjustment. Similar observations have been obtained for CB8 investigated in our previous work [42]. An interesting observation about the absolute values of error estimates is the similarity of the energy RMSEs for the two FM-B97-3c parameter sets. The refitted FM-B97-3c parameter set, regardless of the initial guess (the restraint reference), achieves ~3.7 kcal/mol RMSE, which could be the accuracy limit of the GAFF(2) functional form for the current CB7 host. If we further decompose the energy RMSE into the contributions of individual repeating units, we reach the 3.7/7 = 0.5286 kcal/mol energy error for each repeating unit. This value could be compared with the CB8 result reported in our previous work [42]. Despite the differences in reference levels (BLYP-D4 in the previous work and B97-3c in the current work) and the molecule sizes (CB8 in the previous work and CB7 in the current work), the energy error of each repeating unit in CB8, 4.3/8 = 0.5375 kcal/mol, is very close to the current CB7 result 0.5286 kcal/mol. Therefore, it is highly probable that the accuracy limit of energetics for the CB[*n*] family is ~0.53 kcal/mol relative to high-level ab initio calculations.

As this energy scalar does not provide detailed information on the error distribution inside each molecule, we then check the time series of the atom-specific force errors in Appendix A. In our previous work investigating the larger-ring CB8, the repeating units are numbered sequentially, and the atomic forces on heavy atoms are found to be more problematic than hydrogen atoms [42]. Although this is informative, we expect to grab more detailed descriptions of error distributions in the current CB7 investigation. Thus, we separate the contribution of each element in atom numbering. All nitrogen atoms are put at the beginning of the atom list, and all oxygen atoms are placed at the end of the list. More specifically, the first 28 atoms are nitrogen atoms in the seven repeating units, and the last 14 atoms are oxygen in the -C=O portals. The middle ones are mixed carbon and hydrogen atoms sequentially numbered for the seven repeating units, as hydrogen atoms do not have noticeable problems (white dots). It can be clearly identified from the heatmap that the errors in the transferable GAFF(2) lie mostly at the CB7 backbone (nitrogen and carbon atoms), and the errors on the oxygen of -C=O portals are relatively small. Upon force field refitting, the force errors of all heavy atoms are decreased to similar levels. Interestingly, comparing the averaged force errors under the two refitted force fields, we still observe similar values ~12.5 kcal/(mol·Å·atom), which could be the accuracy limit of the functional form of their base force field GAFF(2).

The above investigation of energetics and atomic forces confirms the room for improvement in the transferable parameter set. We then perform unbiased sampling in explicit solvent to probe the dynamic behavior produced by the refitted parameter sets in order to investigate whether the parameter adjustment would incur noticeable variations of the host dynamics. Similar to the GAFF(2) results in Figure 3c, we superpose the 500 ns trajectory obtained with each refitted parameter set in Appendix A. Comparing these structural arrangements with the GAFF(2) results, we do not see any noticeable difference. Therefore, the regularized parameter adjustment, although it betters energetics and atomic forces, does not dramatically alter the host dynamics. A similar conclusion could be reached from the time-series data of Rg shown in Appendix A. Therefore, the host dynamics produced by pre-fitted GAFF(2) are already good enough for the CB7 host, and further parameter adjustments are unnecessary. However, it should be noted that for enlarged rings (e.g., CB8) with increased structural flexibility, betterments could still be pursued, as suggested in references [42]. Finally, as the GAFF2 description has been shown to be closer to ab initio calculations in our previous investigation of the enlarged ring CB8, for a consistency recommendation for the whole cucurbiturils family, we believe GAFF2 to be a preferred option in practical applications.

#### 3.1.3. Is GAFF2 Good Enough for Further Enlarged Cucurbiturils?

Based on the above-detailed investigation of commonly encountered cucurbiturils, including CB6, CB7, and CB8, the above conclusion on the superiority of GAFF2 over GAFF and the closer-to-higher-level behavior of GAFF2 seems solid enough. However, as inferred in the above discussion, more complex dynamic behaviors could be expected for further enlarged rings. The most popular specie with such a property should be CB10, which features 10 repeating units forming its large macrocycle. We similarly perform 500 ns explicit-solvent sampling of this macrocycle under GAFF and GAFF2 and further refit a force field based on the B97-3c statistics (FM-B97-3c from GAFF and GAFF2). The Rg time series is presented in Figure 4a, where indeed, more complex host dynamics are observed. While the GAFF2 parameter set still seems stiffer than GAFF and maintains a larger Rg at the beginning of the brute-force simulation, both GAFF derivatives tend to stay at the smaller Rg state in most of the sampling time. By contrast, the two FM-B97-3c parameter sets consistently keep the enlarged CB10 ring in its large Rg state, which differs from both transferable force fields. According to structural visualizations, the large Rg state features a single large host cavity widely open to external guests (including water), while in the small Rg state, the host experiences significant conformational changes (similar to the collapsed state of CB8), and the large cavity is split into several (two or three) smaller guest coordination regions. In order to better illustrate the structural variations, we present structural overlays under all of the four parameter sets in Figure 4b and the illustration of the cavity fluctuation in Figure 4c. The two FM-B97-3c parameter sets maintain the largely opened state, the GAFF2 parameter set fluctuates between the large single-cavity state and the slightly collapsed two-cavity state, and the GAFF parameter set is only energetically favorable in the collapsed states with two or three smaller guest-coordinating regions. These observations suggest that the stiffness of the CB10 host cavity follows FM-B97-3c > GAFF2 > GAFF. While the GAFF2-stiffer-than-GAFF phenomenon is expected according to previous investigations of CB8, such GAFF2-different-from-FM behavior is never observed in existing CB*n* (*n* = 6, 7, 8) investigations. As the refitted molecule-specific FM-B97-3c parameter sets are expected to be more accurate than transferable selections, and the two FM-B97-3c parameter sets produce consistent dynamic behaviors of the CB10 host, we reach the conclusion that even GAFF2 is not satisfactory enough. As none of the existing general-purpose force fields is able to produce the correct host dynamics for enlarged cucurbiturils, it seems necessary to refit existing force fields in the molecular modeling of large rings.

### 3.2. Pillararenes

The second host family we study is pillar[*n*]arene. Unlike the previous CB[*n*] family that is better studied and applied in much pharmaceutical research, pillararenes are relatively unpopular and seldom studied in a detailed manner computationally. The modeling reports on host–guest binding affinities in the recent SAMPL9 challenge serve as a good addition to the literature [12,50,51,52], but the dynamic behaviors of this host family are still to be revealed. Due to the low water-solubility of the prototypical form, pillararenes are mostly applied with chemical modifications. For instance, the carboxylated form featuring -CH_2_COO^−^ has enhanced solubility, strong drug-binding ability, and easy functionalization and thus serves as a promising candidate in drug delivery [53,54,55,56,57]. As this modification has been covered in our recent end-point series on the SAMPL9 dataset involving the 6-membered species [50,51,52], in the current work, we consider another chemically modified pillararene series with -SO_3_^2−^ tails on the symmetrical rims. These host series are thus named SP*n* in the following discussion. The 6- and 7-unit members have intermediate cavity volume and entrance size, although the 5-unit macrocycle also has the ability to encapsulate drug-like molecules. 

#### 3.2.1. Term-by-Term Comparison and Host Dynamics under Transferable Parameter Sets

Similar to the first CB[*n*] case, we start with a detailed comparison of force field definitions under the two GAFF versions. The most critical parts for intra-molecular conformational preference, the torsional potentials, are still compared in the first place. The small -SO_3_^−^ tails do not involve complex torsional interactions, and all torsional potentials defined for this SP*n* family are presented in Figure 5a. Through a by-term comparison, we confirm that all torsional parameters defined in GAFF and GAFF2 are the same. Thus, the difference between host dynamics produced by the two GAFF versions, if any, would be caused by the differences in the other force field parameters, including bond stretching, angle bending, and vdW terms. We then extract all bond-stretching and angle-bending terms for this host family in Figure 5b–e. For the bond-stretching potentials, the equilibrium lengths remain unchanged, but the force constants of the three terms in Figure 5b are decreased, while those of the four terms in Figure 5c are increased, especially for the sulfur–oxygen bonds (increases by ~40%). The angle-bending terms exhibit similar behavior, i.e., the terms in Figure 5d,e have minorly increased force constants, while the force constants of those in Figure 5f are increased significantly (by ~50%). As for vdW parameters, there are major updates of both length and energy parameters when shifting from GAFF to GAFF2, similar to the cucurbiturils discussed in the previous section. Overall, bond stretching, angle bending, and vdW interactions within the pillararene host are different to some extent between the two GAFF versions. However, as these differing terms are often believed to have relatively minor influences on intra-molecular conformational preference, it is difficult to reach any conclusive diagnosis of differences between host dynamics. Thus, direct simulations of the host family seem necessary to depict the real picture of dynamic behaviors.

Similar to the cucurbiturils case, we conduct 500 ns explicit-solvent sampling for the three hosts in this sulfur-substituted pillararene family with explicit sodium counterions neutralizing the net charges. Thus, the host cavity could be occupied by water or counterions. The structural overlay under the two GAFF derivatives is presented in Figure 6a. Comparing the host dynamics under GAFF and GAFF2 for individual host molecules, no noticeable difference is identified. The comparison of the Rg time series shown in Figure 6b gives a similar conclusion. Thus, the identified differences in bond stretching, angle bending, and vdW terms do not incur significant alterations in host dynamics.

Aside from the GAFF-vs-GAFF2 comparison, another interesting variation is the size of the host ring (the number of repeating units). For the smallest ring SP5, the host dynamics is rather regular/stiff. The rotation of the -SO_3_^2−^ rims is limited to a small region, and the host backbone fluctuates minimally. When an additional repeating unit is added (SP6), the host dynamics become more flexible. Both the pillararene backbone and the -SO_3_^2−^ rims become noisier on the Figure 6a structural superposition. When the number of repeating units is further increased to 7 (i.e., SP7), the fluctuating behaviors of all regions are magnified. This fluctuation-increasing trend can also be identified from the Rg time series in Figure 6b, where the SP7 curves are obviously nosier than the other. Overall, the ring-size-dependent behavior of the SP*n* family is similar to the cucurbiturils, i.e., larger rings with nosier dynamics. This phenomenon is also expected to be generalizable to other host families, such as cyclodextrins (α-, β-, and γ-forms).

A noteworthy phenomenon in the Rg time series is that for SP*n,* there is no multi-state behavior, and only a single conformational state is observed. The host cavity is not collapsed like the GAFF CB8 in Figure 3c, hinting at differences between conformational fluctuations in different host families. According to the snapshots extracted in Figure 6c, many unexpected host conformations are explored. For example, the -SO_3_^2−^ rims could occupy the entrance of the host cavity or fluctuate away from the cavity and become fully solvent-exposed. The aromatic rings could stay in their well-shaped conformation (see, e.g., 3D chemical structures in Figure 1) or rotate to be perpendicular to the SP7 normal. The stabilizing effects for these unexpected twisted conformations, according to the above force field analyses, include the intrinsic flexibility of the host (ring size) and the force field definitions. Aside from these intra-molecular factors, an additional inter-molecular component also plays a critical role. Specifically, according to the explicit host–ion interactions presented in Figure 6d, the additional inter-molecular stabilizing factor is recognized as the coordination between the host and the cavity-filling ions.

#### 3.2.2. Refitting and Evaluation

The above-detailed GAFF-vs-GAFF2 comparison suggests that the two general-purpose force fields, although having different bond stretching, angle bending, and vdW parameters, produce almost the same dynamic behavior for the SP*n* host family. Thus, practically using GAFF or GAFF2 seems indifferent. However, strictly speaking, one of them must produce closer-to-reality descriptions compared with the other. Therefore, we then perform ab initio calculations, refit each GAFF version and then evaluate the closeness of each force field to the high-level reference. As we include L2 regularization in the loss function, the refitting is initiated with both GAFF and GAFF2, similar to the cucurbiturils situation. The refitting and evaluation follow exactly the detailed workflow presented at the beginning of Section 3, and thus we turn to the numerical results directly. The energetics comparison for GAFF, GAFF2, and the two FM-B97-3c force fields refitted from GAFF and GAFF2 is presented in Appendix A. The huge differences between RMSE/MAE values before and after refitting clearly suggest that the energetics reproduction could be improved dramatically upon GAFF(2) refitting. However, the magnitude of improvements ~10 kcal/mol seems quite unusual according to our previous experiences with macrocyclic hosts [7,42]. To obtain a better understanding of the source of betterments and errors, we monitor the time series of the atom-specific force errors in Appendix A. From the force RMSE given at the top of each atom-detailed heat map, we still observe pronounced improvements upon force field refitting. The six 21-atom repeating units of SP6 are numbered sequentially. The first 11 atoms form the pillararene backbone, while the other 12–21 atoms are the two -OSO_3_^−^ rims at both entrances. From the heatmap, it can be clearly identified that the improvements in refitting happen at the -SO_4_^−^ rims. By further comparing the term-specific parameters in the refitted FM-B97-3c and the initial guess, we identify that the S-O bond-stretching term marked in Figure 5c experiences the largest variations upon refitting. Regardless of the initial guess, both its equilibrium length and force constant are altered significantly upon refitting, e.g., bond length varying by ~7%. Therefore, the pronounced betterments in refitting are caused by problematic -SO_4_^−^ parameters (especially the S-O linkage) in general-purpose force fields. As for the GAFF-vs-GAFF2 comparison, the energetic and force data (RMSE) suggest that GAFF is slightly closer to the high-level reference than GAFF2. As both GAFF and GAFF2 provide similar dynamic behaviors, energetically speaking, GAFF is superior to GAFF2.

The above numerical data of energetics and atomic forces provide valuable information about the closeness of transferable parameter sets to high-level reference and successfully locate the problematic terms. We then perform 500 ns unbiased explicit-solvent sampling of the solvated host under the refitted parameter sets in order to probe whether the pronounced energetic and force betterments would lead to noticeable differences in host dynamics. The host structures are superposed in Appendix A for the two FM-B97-3c force fields refitted from GAFF and GAFF2. By comparing the structural overlays of the two refitted force fields, we do not identify significant differences. Likewise, the FM-B97-3c dynamics in Appendix A are also similar to those under the initial guesses GAFF(2) in Figure 6a. The time series of the radius of gyration and the refitted force fields are also compared with those under the initial guesses in Appendix A, where the four parameter sets share similar behaviors. Therefore, despite the significant problems in bond-stretching potentials (and also minor problems in angle-bending and vdW terms), the dynamic behavior of the sulfur-substituted pillararene family could be accurately described with transferable GAFF derivatives. Practically, no significant difference exists between host dynamics produced by GAFF and GAFF2, and picking either option seems fine, although energetically, GAFF is slightly better than GAFF2. However, it should be noted that when bond-length-related observables or in situations where accurate energetics are crucial (e.g., reweighting between force field descriptions and ab initio levels) [58,59,60,61], the pre-fitted GAFF derivatives could be non-satisfactory and refitted force fields could be necessary.

### 3.3. Cyclodextrins

Cyclodextrins are probably the most commonly applied macrocycles due to their readily availability, low cost, and tunable solubility [62,63,64]. Their glucose units have many donors and acceptors for hydrogen bonds and hold a hydrophobic cavity encapsulating drug-like molecules. Among the bio-safe species with 6–8 repeating units (i.e., α-, β- and γ-CD forms), the 7-unit β-CD is of the highest popularity due to the compatibility of its portal size with most pharmacologically active molecules [65,66,67,68]. In practical applications, the prototype CD does not often fulfill the intended usage, and chemical modifications are introduced to fine-tune the physiochemical properties of the macrocyclic container. As the ring-size dependence has been illustrated extensively in the previous cucurbituril and pillararene cases and the prototypical β-CD has been studied in our recent work [7], here in cyclodextrins, we would focus on a methylated β-CD derivative named Hβ-CD as a critical example of chemically modified species. As shown in Figure 1, the methylations of -OH groups result in variations in entrance size and hydrogen-bonding ability and also an increase in conformational flexibility.

#### 3.3.1. Term-by-Term Comparison between Transferable Parameter Sets

In existing studies involving a comparison between GAFF derivatives for the CD family, the focused investigation targets are mostly binding thermodynamics between host–guest complexes and sometimes the dynamic behaviors produced by different parameter sets. To our best knowledge, a detailed comparison at a term-specific level was never presented up to now. Thus, the first step of our force field investigation fills this critical gap. As the differences in bond stretching, angle bending, and vdW terms have been shown to have small impacts on host dynamics in the pillararene investigation presented in the previous section, we thus focus on only the torsional terms. Unlike pillararenes and cucurbiturils, having only several torsional terms regulating the dynamics of the macrocyclic cavity, we identify a larger number (~20) of torsional terms in the CD hosts. According to the presence/absence and types of variations between the two GAFF derivatives, the torsional potentials in Hβ-CD could be divided into five groups, all of which are depicted in Figure 7. The four terms in the first group shown in Figure 7a are defined in both GAFF versions, but their phases and barrier heights are slightly different between the two GAFF derivatives. The two terms from the second group shown in Figure 7b are defined in both GAFF and GAFF2. However, each term is defined by two sub-potentials with periodicities 2 and 3 in GAFF, while an additional torsional potential with the periodicity of 1 is added in GAFF2. Even for the sub-potentials with the same periodicity (i.e., 2 and 3), the height of the barrier and the phase are altered. Thus, these two terms exhibit slightly more differences compared with the first group. The third group contains only a single C-C-C-H dihedral, shown in Figure 7c. Only for this torsional term, the GAFF and GAFF2 definitions are exactly the same. The fourth group also contains only one dihedral formed between C, C, O, and H atoms, as shown in Figure 7d. This torsional term is only defined in GAFF and is eliminated in GAFF2. The four terms in the fifth group shown in Figure 7e are absent in GAFF and thus are newly defined torsional potentials in GAFF2. Overall, the term-specific comparison reveals that most torsional potentials differ between GAFF derivatives, and only one torsional term remains unchanged in GAFF and GAFF2 (i.e., exactly the same definition). Thus, the CD description in GAFF2 indeed seems like a major update of GAFF and would differ noticeably from GAFF, which agrees with the major differences in the dynamic behavior of the prototype β-CD observed in our previous work [7]. More insights could be obtained by investigating the locations of these differing terms in Figure 7. Specifically, the force field alterations happen mainly at the CD backbone, which would be the location where the GAFF and GAFF2 dynamics differ. This conclusion could be safely generalized to other CD derivatives due to structural similarities. For the methylated positions at both entrances, i.e., -CH_2_-OMe at the primary surface and -OMe at the secondary surface (see more detailed discussions about this surface difference in our previous work) [7], there is no definition of torsional potential in this region under both GAFF versions. Consequently, it is expected to observe flexible rotations of the methyl groups at both primary and secondary surfaces under both GAFF derivatives. Unlike the previous cucurbituril and pillararene cases, here we would proceed to force field refitting and energetic evaluation, after which the host dynamics under different parameter sets would be compared.

#### 3.3.2. Refitting and Evaluation

We then turn to the force field refitting and evaluation part to grab perspectives from energetics and about the closeness to high-level descriptions. The computational protocol is still similar to the previous cases and would not be repeatedly discussed. The energy deviations of the transferable force fields and the refitted parameter sets are presented in Appendix A, where significant improvements could be observed upon force field refitting. The magnitude of error decreases is similar to cucurbiturils and falls within the normal range, and no huge error like SP*n* is observed. Concerning the relative accuracy of GAFF and GAFF2, it is observed that the GAFF2 errors (either RMSE or MAE) are systematically smaller than GAFF. A similar phenomenon is observed for the refitted FM-B97-3c force field that is optimized in the presence of L2 restraints restraining the parameter space in the neighborhood of the initial guess. These phenomena suggest that energetically GAFF2 should be closer to the high-level reference B97-3c than GAFF.

To grab further insights into the sources of errors, we turn to the atom-specific heatmaps of force errors shown in Appendix A. Comparing the force RMSE at the top of each plot, we still observe that GAFF2 and its refitted form systematically outperform the GAFF counterparts. Therefore, GAFF2 outperforms GAFF in both energetics and atomic force and provides a description closer to the high-level reference. Similar conclusions have been reached in our previous work investigating the prototype β-CD [7]. Therefore, this GAFF2-better-than-GAFF should be quite general for the CD family. To grab more detailed insights into the error distributions, we number the carbon atoms of the CD backbone, the other heavy atoms (backbone oxygen and substituting methyl groups), and hydrogen atoms separately. The first 42 atoms are the CD backbone (6 carbon atoms per repeating unit), the No. 43–91 atoms are the other heavy atoms (oxygen and methyl tails), and the 92–189 atoms are hydrogen atoms. An illustration of this atom grouping regime is depicted in the inset of Appendix A. Following this numbering scheme, we identify from the heatmap that the force errors mainly distribute at the backbone carbon atoms under both GAFF versions. The backbone oxygen and methyl substitutions are relatively well-described, and the errors on hydrogen atoms seem negligible. Upon regularized refitting, the errors on heavy atoms are minimized to a similar level, validating the improved accuracy of force descriptions in FM-B97-3c parameter sets.

#### 3.3.3. Host Dynamics Produced by Transferable and Molecule-Specific Refitted Force Fields

Finally, with a detailed understanding of the similarities and differences of transferable GAFF derivatives and the verified higher accuracy and reliability of the refitted FM-B97-3c parameter sets, we compare the host dynamics under pre-fitted general-purpose and refitted molecule-specific force fields. The superpositions of the 500 ns explicit-solvent simulations under the four parameter sets are depicted in Figure 8a. For clarity, the hydrogen atoms are omitted in this presentation, and only the CD backbone and the substituting methyl groups are retained. The Rg time series with representative structures extracted from the GAFF and GAFF2 trajectories are shown in Figure 8b. No counter ion is added in the simulation box for this net-neutral host. Thus, the dynamic behavior is regulated by the intra-molecular interactions within the Hβ-CD host and the inter-molecular host–water interactions.

According to the structural overlay in Figure 8a, the host cavity is twisted and stays in a squashed state under GAFF. This conclusion could be more evident according to the GAFF snapshot with explicit hydrogen presentation at the small Rg configuration presented in Figure 8b. Under GAFF2, the host dynamics are more regular, and the host cavity seems stiffer, which agrees with the observations in the prototype β-CD reported in our previous work [7]. However, the substituting methyl groups are still invading the entrance region with a non-negligible probability, which is somehow similar to GAFF. It should be noted that this conformation (i.e., the occupancy of the cavity entrance by the substituting methyl groups) should be differentiated from the squashed state where the host cavity is squeezed, as the latter is caused solely by backbone motions, while the former involves both backbone motions and the rotational dynamics of the methyl rims. Unlike the transferable parameter sets, the refitted force field FM-B97-3c, regardless of the initial guess and restraint reference, produces an even more regular dynamic behavior compared with GAFF2. The FM-B97-3c host cavity is fully opened during the course of 500 ns unbiased sampling, and the substituting groups seldom visit the entrance region. These two refitted force fields are thus producing the stiffest description of the Hβ-CD cavity in this work. As the refitted force fields are energetically closer to ab initio calculations (higher reliability) and the host dynamics produced by them are initial-guess-independent, their host dynamics should be close-to-ab initio and of the highest reliability. Therefore, none of the popular GAFF derivatives could be satisfactory for the substituted CD rings, and force field refitting with costly ab initio calculations seems necessary to accurately describe the host dynamics. From the Rg time series presented in Figure 8b, the similarity of the refitted force fields could still be observed. The GAFF curve is systematically lower than the other, and the GAFF2 curve is closer to FM-B97-3c results, but still, some deviations could be observed. Thus, similar conclusions could be obtained according to the Rg time series.

Overall, the detailed investigation of the methylated β-CD ring provides insights into the similarities and differences of transferable GAFF derivatives, their closeness to high-level descriptions, their room for improvements, and the dynamic behaviors of pre-fitted and refitted force fields. When accurate descriptions of the host dynamics are pursued, force field refitting seems necessary. However, as this requires costly ab initio calculations if the practitioners insist on employing transferable force fields, the GAFF2 option could be recommended, considering its proven successes in host dynamics and binding thermodynamics for the prototype β-CD reported in our previous work [7] and its close-to-ab-initio behavior for chemically modified β-CD reported in the current work.

### 3.4. OA

The last host family under investigation is the basket-like OA. Unlike other macrocycles investigated in this work, the hosts from the OA family have a single entrance to a deep hydrophobic cavity. Consequently, OA hosts are often recognized as deep-cavity cavitands [69]. The carboxyl groups connected to the alkyl chains and the aromatic rings at the cavity entrance increase the solubility of OA hosts. Chemical modifications are sometimes introduced to adjust the entrance size [70,71,72]. In molecular simulations of OA derivatives, general-purpose force fields such as GAFF derivatives have been widely applied [73,74,75]. Unfortunately, to our best knowledge, no validation benchmark has ever been considered up to now. To this aim, we present an unprecedently detailed investigation of force field behaviors and refitting analyses for the prototype OA.

#### 3.4.1. Term-by-Term Comparison between Transferable Parameter Sets

We still first compare the detailed definitions of torsional terms in GAFF and GAFF2 to grab some preliminary insights into host dynamics without dynamics simulations. Similar to CDs, the OA host has a large number of torsional potentials regulating its dynamic behavior. The GAFF2 definitions of the four terms in Figure 9a are exactly the same as those in GAFF. The five terms in Figure 9b are defined in both GAFF derivatives, but their barrier heights are altered with varying degrees. Specifically, the barrier height of the first term is elevated almost by a factor of 2 in GAFF2, while those of the other four terms are marginally adjusted (either increased or decreased). There are four torsional potentials newly defined in GAFF2, as shown in Figure 9c. Among the differing terms (i.e., Figure 9b,c), the last term in Figure 9b and the first term in Figure 9c are located at the cavity entrance. Compared with GAFF, the former in GAFF2 has a decreased barrier height (from 1 kcal/mol to 0.52 kcal/mol), hinting at slightly higher flexibility in the -COO^−^ region. However, as this height of torsional potential is still as high as the thermal energy (~0.59 kcal/mol at 298 K), the -COO^−^ region is still expected to be rather regular and does not exhibit flexible rotation like the -CH_2_-OMe and -OMe tails in the previous methylated β-CD. The latter strengthens the cavity entrance, but its low barrier height of ~0.9 kcal/mol may not perturb the existing dynamic behavior significantly. The other newly added and differing terms are mainly imposed on the bottom of the host and make this region more regular under GAFF2. Aside from the torsional potentials, the other terms (i.e., bond stretching, angle bending, and vdW parameters) also exhibit differences between GAFF derivatives. However, according to the analysis in the sulfated pillararene, these terms would not incur noticeable alterations in host dynamics. Therefore, according to the similarities and differences identified in the detailed force field analysis, the stiffness of the host entrance and the rotational dynamics in the -COO^−^ region could be altered when shifting from GAFF to GAFF2, but the magnitude of variations could be insignificant.

#### 3.4.2. Refitting and Evaluation

We then proceed to the refitting and evaluation stage for the OA host. The energy deviations of the transferable GAFF derivatives and the refitted force fields are depicted in Appendix A. The decrease of energetic RMSE is within the reasonable range and comparable to cucurbiturils and cyclodextrins, and no significant improvement, like the pillararene case, is observed. The error sizes of the GAFF derivatives are similar, and a similar phenomenon is observed for the two refitted force fields. Thus, the closeness of GAFF to ab initio reference, energetically speaking, is similar to the GAFF2 counterpart. The by-atom heatmaps shown in Appendix A provide more detailed information. We number all heavy atoms first (No. 1–128) and then the hydrogen atoms (No. 129–184). From the time series of force errors, we know that atomic forces on heavy atoms are more problematic under both GAFF derivatives, while the force field refitting leads to much better force descriptions. The force RMSE values under the two GAFF derivatives are very similar, ~22.4 kcal/(mol·Å·atom), and those of the refitted force fields are ~14.3 kcal/(mol·Å·atom), which suggests the similarity of accuracies of the two transferable force fields. Overall, energetically we cannot conclude whether one GAFF derivative outperforms the other.

#### 3.4.3. Host Dynamics Produced by Transferable and Molecule-Specific Refitted Force Fields

Although the by-term force field comparison identifies differences between GAFF derivatives, the two transferable force fields do not have noticeable distinctions for energetic and force metrics. Thus, we expect to grab more insights from the dynamic viewpoint presented here. Still, 500 ns unbiased sampling is performed under each parameter set, and the structural overlap and the Rg time series are presented in Figure 10a,b, respectively. According to the structural superposition, the host dynamics produced by transferable GAFF derivatives and refitted FM-B97-3c force fields are very similar. Under all of the four parameter sets, the cavity-holding region is rather regular. However, some minor conformational fluctuations are observed for the benzoic acid group(s) at the cavity entrance under GAFF derivatives. Specifically, some of the four benzoic acid groups could fluctuate between the ordinary down conformation and a newly observed up state, although the latter seems energetically less favorable and is seldom visited. To further elucidate the flipping of the benzoic acid groups, we visualize the trajectories and present the relevant snapshots in Figure 10c,d, where this up–down transformation could happen with and without host–ion coordination. Therefore, this up–down rotational dynamics involves the interplay of the torsional potentials of the host itself and the inter-molecular host–ion coordination. Unlike GAFF derivatives, the benzoic acid groups under the two refitted force fields are more regular and never visit the up conformation. The Rg time series under the four parameter sets are extremely similar and exhibit equilibrium fluctuating behaviors, from which we conclude that the host dynamics under the four parameter sets are similar. The up–down flipping of the benzoic acid groups cannot be clearly identified in the Rg time series, which reflects the insensitivity of this observable to some extent. Overall, generally, the two GAFF derivatives behave similarly to the refitted high-accuracy FM-B97-3c force fields, although the benzoic acid groups at the cavity entrance sometimes exhibit minor conformational flipping. If the practitioners prefer general-purpose force fields, both GAFF derivatives could be applied for OA hosts, as this would not cause significant trouble. For users preferring high-accuracy force field descriptions, additional force field refitting could be attempted.

## 4. Concluding Remarks

Accurate modeling of host–guest systems remains rather challenging in the modern computational community. One of the main influencing and limiting factors is the force field accuracy. Despite the massive application of general-purpose force fields in the molecular modeling of host–guest complexes, the accuracy levels and dynamic behaviors under these transferable parameter sets are not well-understood. To answer this long-standing question and consolidate modeling protocols of host–guest systems, we devote the current paper to an unprecedently detailed investigation of commonly employed fixed-charge force fields.

The first force field component that is believed to exhibit significant system dependencies in host–guest modeling is the atomic charge. Common practices are semi-empirical models (e.g., AM1-BCC) and the rigorous regularized least-squares RESP regime with different levels for ESP scans. Comparison between ESP RRMSE suggests that AM1-BCC models produce acceptable results in some cases, but the RESP regime serves as a fail-safe option that is recommended in general, as long as sufficient computational resources are available. It is often argued that the traditional HF/6-31G* selection is too old and crude to be applied in modern modeling studies. However, our detailed validation based on a huge basis set def2-QZVPP (practically converged in charge generation and also condensed-phase simulations) and a scan of modern DFT selections supports the opposite conclusion. There is no solid numerical evidence supporting the shift to larger basis sets and higher and costlier ab initio levels. Although we would not fully discommend the use of DFT levels, it should be emphasized that the traditional HF/6-31G* still serves as an acceptable option in charge generation and should not be criticized in practical applications.

The other force field components, including bond stretching, angle bending, torsional, vdW, and many other related parameters, are predominantly extracted from transferable force fields, especially GAFF derivatives. These parameters are commonly believed to exhibit small system-dependent behaviors. However, applications without justification are rather dangerous, and detailed force field validations should be properly conducted. Lucky shots due to fortuitous and possibly erroneous error cancellations are poisonous to this field. The obtained structural and thermodynamic insights could be biased, and the reliability of the host–guest interaction picture would be questionable. Therefore, the main results of the current paper focus on detailed comparison, validation, and refitting of transferable force fields, in order to identify their similarities and differences, their deviations from ab initio references, and their room for improvement.

The first family under investigation, cucurbiturils, is relatively well-studied, with many publications from both experimental and computational sides. The pumpkin-like family CB*n* is formed by *n* glycoluril monomers linked by 2*n* methylene bridges. The hydrophobic cavity encapsulates external species, while the hydrophilic carbonyl portals act as hydrogen-bond acceptors. The 6-, 7-, and 8-unit species (especially the latter two) are popular in practical applications due to the similarity of their portal sizes and pharmaceutically active molecules. By comparing the detailed force field terms of GAFF and GAFF2, we identify two torsional potentials with varied definitions under GAFF derivatives. As these terms are directly acting on the host backbone, the stiffness of the host ring is expected to exhibit a force field-dependent behavior. However, practical simulations in explicit solvent reveal that only the CB8 dynamics exhibit noticeable differences between the two GAFF derivatives, while the structural overlays under GAFF and GAFF2 are virtually the same for the other two species. This phenomenon suggests that the host dynamics are not only determined by the force field definitions and are also related to the intrinsic flexibility of the host ring. Only when the ring size is increased to a sufficient level the force field differences could finally impact the dynamic behavior of the host cavity. In energetics evaluation and parameter refitting, the 7-unit member is selected as the investigation target, as the other (i.e., CB8) has been studied thoroughly in our previous work [42]. It is observed that the energy error of the refitted force field, regardless of the initial guess (also the restraint reference), is ~3.7 kcal/mol for the whole CB7 molecule and ~0.53 kcal/mol for each repeating unit, which could be the accuracy limit of the given functional form of the force field. Similarly, for the force error, the accuracy limit is ~12.5 kcal/(mol·Å·atom). The atom-specific heatmap of force errors further locates the CB7 backbone (nitrogen and carbon) as the main problematic atoms, while the C=O portals and hydrogen atoms are relatively well-described by transferable force fields. The refitted force fields produce host dynamics similar to transferable force fields. Thus, when high-accuracy energetics are not pursued, transferable force fields are still applicable. According to our previous work, the GAFF2 description for CB8 is closer to ab initio calculations than GAFF. Therefore, for a consistency recommendation, we believe GAFF2 to be the recommended option for the popular CB6, CB7, and CB8 species in the CB*n* family. More complex dynamics are observed for further enlarged rings; for enlarged cucurbiturils such as CB10, the flaw/inaccuracy of the GAFF2 parameter set begins to produce noticeable deviations from the FM-B97-3c parameter sets. The host dynamics under all existing general-purpose force fields do not agree well with the higher-level description. Thus, the refitted force field as the ultimate fail-safe treatment is necessary for accurate host dynamics in enlarged cucurbiturils.

The pillararene hosts, as another crucial family of macrocycles, are relatively unpopular, and their dynamic behaviors are not well-understood. Due to the unsatisfactory physiochemical properties of the prototype pillararenes (e.g., low solubility) and the size comparability of the 6-unit species, we pick the 6-unit pillararene backbone and consider the sulfur-substituted form SP6 as the investigation target. By-term force field investigation reveals that the sulfur-substituted pillararene family SP*n* shares the same torsional description under GAFF and GAFF2. Thus, the investigation of the SP6 host distills the impact of variations of bond stretching, angle bending, and vdW parameters. Despite the differences in these terms, in practical explicit-solvent simulations, the host dynamics are not altered noticeably. The ring-size-dependent behavior in this pillararene family is similar to cucurbiturils, i.e., with the increase in the size of the host ring, the intrinsic flexibility is increased, and the host dynamics become more complex. A difference between pillararenes and cucurbiturils is that in SP*n,* the host dynamics do not exhibit a multi-state behavior for the largest component of the host family. The host conformation becomes twisted in enlarged rings, but there is no explicit barrier between different conformations. The stabilizing factors for the twisted conformations have two origins. The intra-molecular components include the intrinsic flexibility of the host cavity (ring size) and the force field-defined behavior, while the inter-molecular component is the coordination between the host and the sodium cations. As for the energetics perspective of the force field quality, we observe unusually betterments upon force field refitting. Detailed analyses reveal that the significant improvements are closely related to the problematic parameters in the -OSO_3_^2-^ rims, especially the bond-stretching potentials of the O-S linkage. However, despite the huge energetic biases due to the bond-stretching errors, the host dynamics produced by transferable force fields and the accurately refitted FM-B97-3c parameter sets are virtually identical. Thus, applying either GAFF or GAFF2 would not trigger huge problems in practical applications. However, in cases where bond-stretching motions are crucial, or the reproduction of these energetics is desirable, refitting to secure an accurate description is still preferable.

The third host family studied is CD. This is probably the most popular family with wide applications due to their readily availability and tunability. The prototype 7-unit β-CD has been studied extensively in our previous work, where a complex multi-modal binding behavior is observed for CD-guest complexes, and the GAFF2 parameter set prevails among GAFF derivatives. However, as the prototype CDs do not often satisfy the intended use, chemical modifications are often introduced. Thus, it is preferable to have some detailed yet general insights into the dynamics of these chemically modified rings. The methylated Hβ-CD featuring the drug-compatible 7-unit portal size and chemical modification (methyl substitutions) serves as a good example illustrating the basic features of commonly applied CD derivatives. The unprecedently detailed by-term force field investigation identifies a large number of torsional potentials, most of which differ between GAFF derivatives. These differing terms are all acting on the CD backbone, while no torsional potential is imposed on substituting methyl groups. Thus, the backbone motions under GAFF derivatives are expected to exhibit differences, while the rotational dynamics of the methyl rims should be similar. Regularized force field refitting leads to closer-to-ab initio descriptions. In terms of both system energy and atomic force, the GAFF2 error is smaller than GAFF, and a similar phenomenon is observed for the refitted force fields. Thus, generally, GAFF2 is indeed better than GAFF. As similar conclusions have been reached in our previous investigation of the prototype β-CD, we expect the superiority of GAFF2 would be rather general for the CD family. However, when monitoring the host dynamics produced by the pre-fitted GAFF derivatives and the refitted FM-B97-3c parameter sets, we observe that neither GAFF nor GAFF2 provides an accurate description of the host dynamics. The GAFF cavity remains squashed in unbiased explicit-solvent sampling. The GAFF2 ring seems stiffer, but the cavity entrance is sometimes occupied by the methyl rims. In stark contrast, the host dynamics under FM-B97-3c exhibit an initial-guess-independent behavior, and the host cavity remains fully open with few penetrations by host rims. Therefore, when accurate descriptions of host dynamics are expected, the refitting of CD force fields seems necessary. For practitioners who lack resources to support costly ab initio calculations needed in force field refitting, we believe the GAFF2 option to be a workable solution, considering the proven success in binding thermodynamics and host dynamics for the prototype β-CD reported in our previous work [7]. However, the defects, pitfalls, and potential biases in the simulation outcome should be properly acknowledged.

The last host family under investigation, the OA family, is structurally dissimilar to the previous three. The bowl-like OA host has only a single entrance to a deep hydrophobic cavity. Different chemical modifications could be introduced to regulate the entrance size. Similar to CD hosts, still a large number of torsional potentials could be identified under GAFF derivatives. For some torsional terms, the GAFF and GAFF2 definitions are exactly the same, while for others, the force field definitions are varied (added, eliminated, or changed) between GAFF derivatives. However, the magnitudes of variations are often small, and thus no significant alteration of host dynamics is expected. When comparing the GAFF(2) energetics and atomic forces with ab initio calculations, the deviations from high-level references are similar for the two GAFF versions. A similar phenomenon is observed for refitted force fields. Thus, GAFF and GAFF2 are comparable in accuracy. Practical simulations under pre-fitted transferable and refitted molecule-specific force fields reveal minor differences between host dynamics produced by GAFF derivatives and FM-B97-3c force fields. Specifically, GAFF derivatives are similarly describing a stiff host backbone with occasional up–down flipping of benzoic acid groups at the cavity entrance, which involves the interplay of host–ion coordination and intra-molecular torsional potentials. The stiffness of the host backbone under the high-accuracy FM-B97-3c parameter sets is similar to the GAFF(2) description, but the dynamics of benzoic acid groups become more regular with negligible visits of the unexpected up conformation. Overall, when force field refitting is impractical (e.g., due to lack of computing time), both GAFF and GAFF2 could be applied for the OA family. When high-accuracy energetics and host dynamics are pursued, still, refitted force fields are preferred.

## Figures and Tables

**Figure 1 molecules-28-05940-f001:**
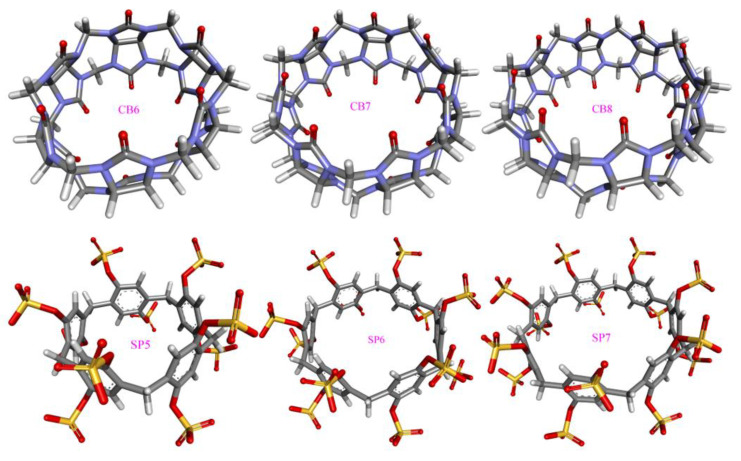
3D chemical structures of macrocycles investigated in the current work.

**Figure 2 molecules-28-05940-f002:**
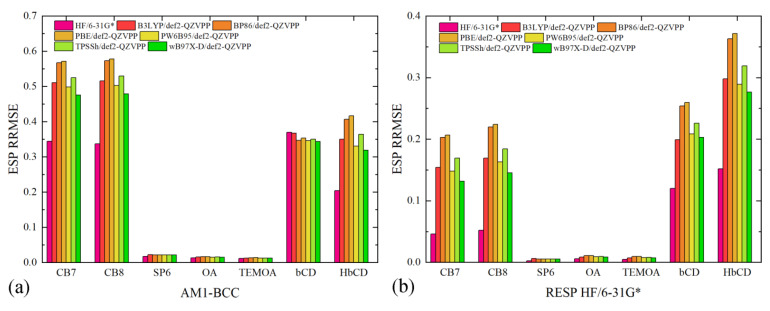
The percentage errors of charge-produced ESP from different ab initio references for macrocyclic hosts with (**a**) AM1-BCC charges, (**b**) HF/6-31G*-targeted RESP charges, and (**c**) PW6B95/def2-QZVPP-generated RESP charges. The ESP deviations for OA, TEMOA, and SP6 are quite small, while noticeable deviations exist for CB7, CB8, β-CD, and Hβ-CD. The semi-empirical AM1-BCC charge scheme produces the largest ESP RRMSEs among the three. The HF/6-31G*-targeted RESP charges have small errors compared with its fitting target, but when compared with the other higher-level selections, the ESP RRMSEs are quite large. For the PW6B95/def2-QZVPP-targeted RESP charges, the ESP deviations from all ab initio references are smaller than the other two charge schemes. The deviations from the traditional HF/6-31G* level are larger than those from the costly def2-QZVPP ones in most situations.

**Figure 3 molecules-28-05940-f003:**
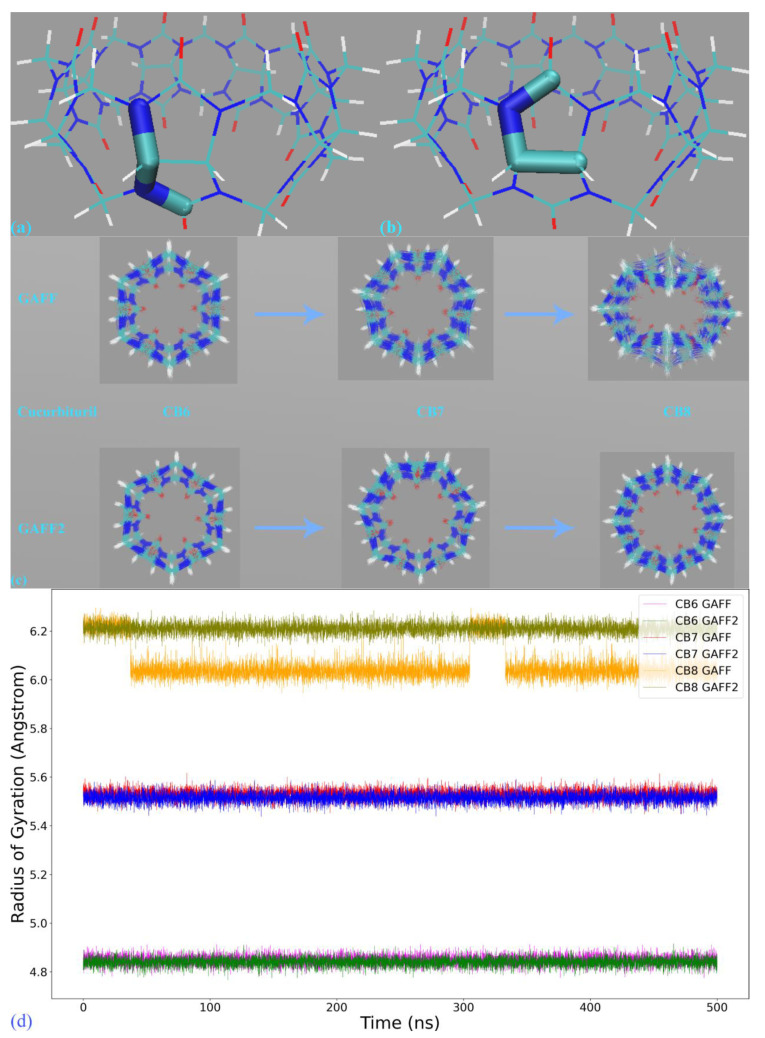
(**a**,**b**) Torsional terms that exhibit noticeable differences in the two GAFF versions in CB7. The (**a**) torsional term is not defined in GAFF, while in GAFF2, it receives an explicit ~2.08 kcal/mol barrier definition. The (**b**) torsional term is defined in both GAFF derivatives, but the phases and barrier heights are altered when shifting from GAFF to GAFF2. (**c**) Superpositions (5 ns per snapshot) of the host structure during the 500 ns explicit-solvent sampling of the solvated host under GAFF and GAFF2 for the CB[*n*] family with varying numbers of repeating units. Despite the different stiffness defined in GAFF and GAFF2, for small rings with six and seven repeating units (i.e., CB6 and CB7), the host dynamics are generally the same under the two general-purpose force fields. When the number of repeating units increases further and hits eight (i.e., CB8), the differences in torsional terms begin to trigger noticeable differences in host dynamics, suggesting an interplay between the torsional potentials and the ring flexibility. (**d**) Time series of the radius of gyration (Rg) of the host molecules (sampling interval 50 ps). For the two smaller rings, CB6 and CB7, there is no significant difference between GAFF and GAFF2 results. When it turns to the large and more flexible CB8 ring, under GAFF, there are two conformational states, with the larger-Rg state featuring opened cavity and the other with a closed/squashed central cavity. By contrast, under GAFF2, the host remains in the opened state during the course of the 500 ns explicit-solvent sampling.

**Figure 4 molecules-28-05940-f004:**
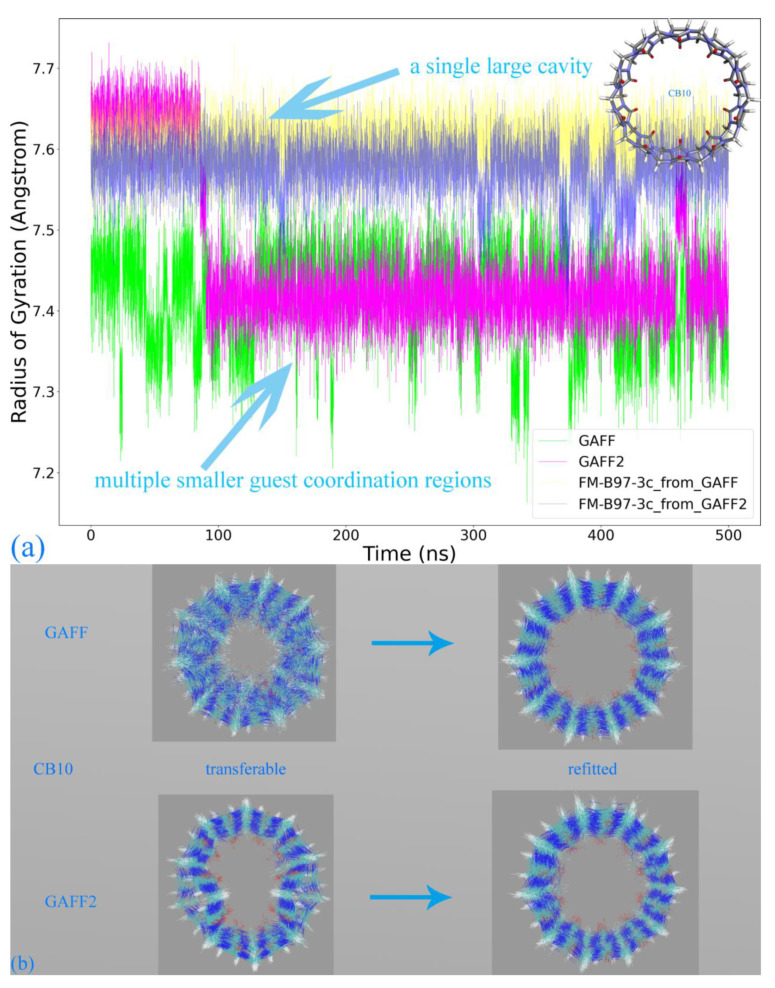
For the further enlarged ring CB10, (**a**) the time series of Rg under GAFF derivatives and the refitted FM force fields, (**b**) structural overlay, and (**c**) the three conformations observed for the host cavity with marks indicating the force field favoring them.

**Figure 5 molecules-28-05940-f005:**
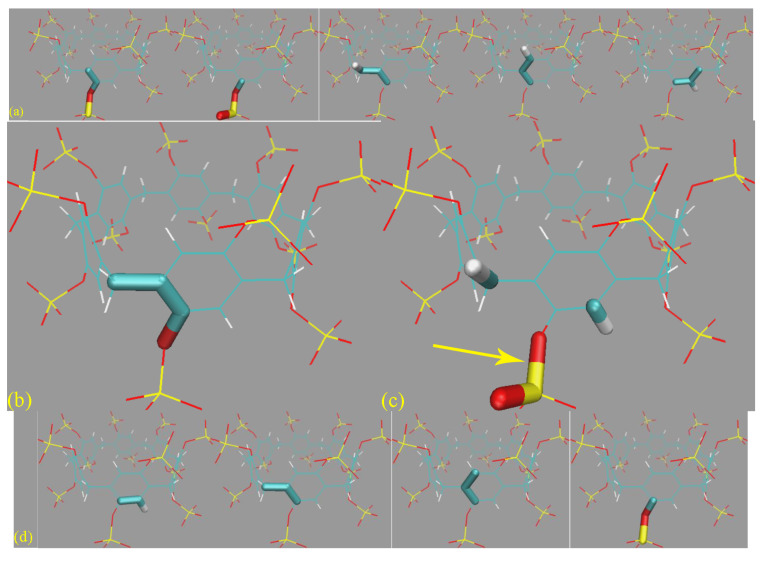
(**a**) Torsional terms for the SP6 host. The parameters for all torsional potentials are the same under the two GAFF derivatives. (**b**,**c**) Bond-stretching terms involved in the SP6 host. The equilibrium lengths of these harmonic bond-stretching potentials remain unchanged when shifting from GAFF to GAFF2, but the force constants are altered with varying degrees. The force constants of the three terms in the subplot (**b**) are decreased, while those of the four terms in (**c**) are increased, especially for sulfur–oxygen bonds in the -SO_3_^−^ group (increases by ~40%). (**d**,**e**) Angle-bending potentials that are similar in the two GAFF versions (with minorly increased force constants, to be specific) and (**f**) those that exhibit significant differences between GAFF and GAFF2 (force constants increase by ~50%). The marked bond-stretching term (the O-S linkage) experiences the largest variation upon FM refitting, regardless of the initial guess (i.e., GAFF or GAFF2).

**Figure 6 molecules-28-05940-f006:**
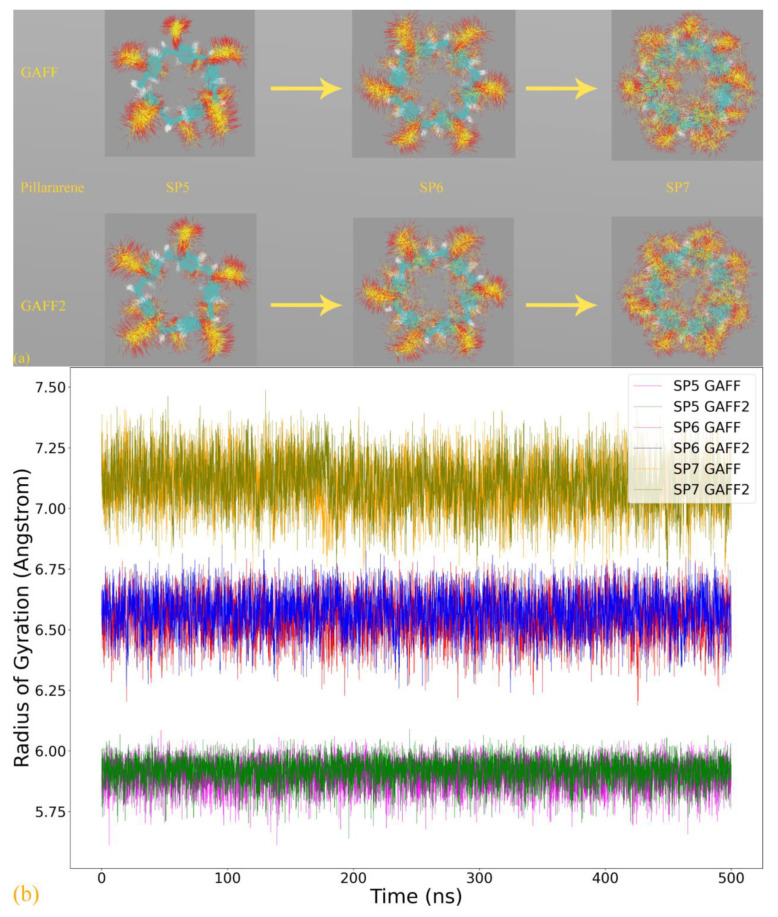
(**a**) Superpositions (5 ns per snapshot) of the host structure and (**b**) the time series of Rg during the 500 ns explicit-solvent sampling of the solvated host under GAFF and GAFF2 for the sulfur-substituted pillararene family with a varying number of repeating units. The 5- and 6-membered rings are obviously more regular than the 7-membered host, and their cavities remain fully open during the course of the 500 ns unbiased simulation. By contrast, the enlarged ring SP7 exhibits higher structural flexibility and explores many unexpected conformations (e.g., those shown in the subplot (**c**)), which agrees with the trend in the CB[*n*] family observed in the previous section. In subplot (**d**), the host–ion interactions are explicitly shown, which provides hints on favorable interactions when the host molecule stays in a twisted conformation. The fluctuation amplification can also be observed in the Rg time series, where the SP7 curves exhibit larger fluctuations compared with SP5 and SP6. The host dynamics produced by the two GAFF versions do not exhibit significant differences, which indicates that the variations in bond stretching, angle bending, and vdW parameters collectively have little impact on the practical dynamic behaviors.

**Figure 7 molecules-28-05940-f007:**
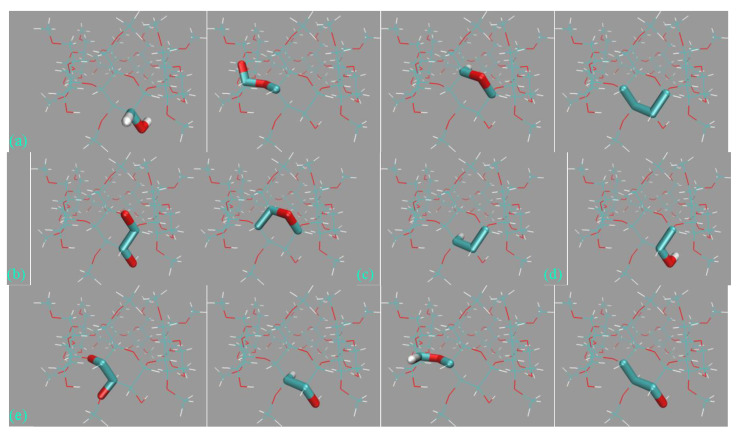
In the methylated β-CD, (**a**) torsional terms defined in both GAFF versions but with different phases and barrier heights, (**b**) those with different phases, barrier heights, and also an additional term with the periodicity of 1 newly defined in GAFF2, (**c**) exactly the same in the two GAFF versions, (**d**) eliminated in GAFF2 (i.e., only presented in GAFF), and (**e**) newly defined in GAFF2. Interestingly, there is no torsional potential imposed on -CH_2_-OMe and -OMe groups. Also, considering the fact that the differing terms locate at the CD backbone, the backbone dynamics of CD derivatives are expected to exhibit differences between GAFF derivatives, but the rotational dynamics of the -CH_2_-OMe and -OMe tails (substitution tails) should be the GAFF-version-independent.

**Figure 8 molecules-28-05940-f008:**
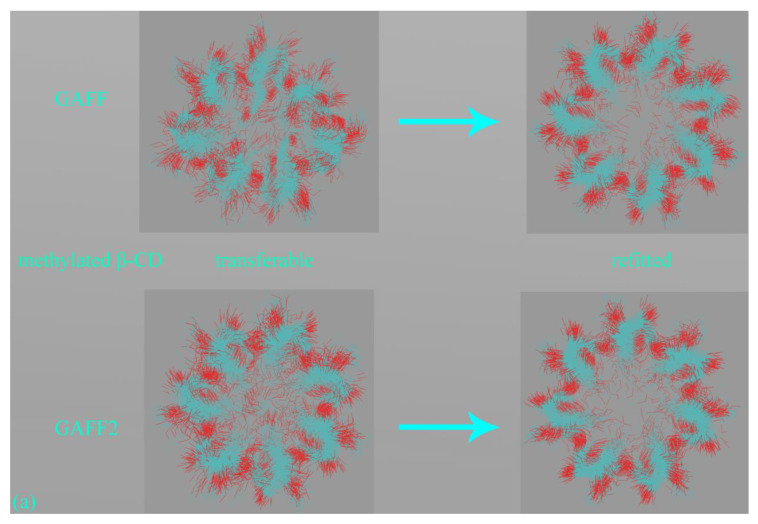
(**a**) Superpositions (5 ns per snapshot) of the host structure during the 500 ns explicit-solvent sampling of the solvated host under transferable and refitted force fields for the methylated β-CD. (**b**) Time series of the radius of gyration (Rg) of the host molecules (sampling interval 50 ps) with representative snapshots extracted from GAFF and GAFF2 trajectories.

**Figure 9 molecules-28-05940-f009:**
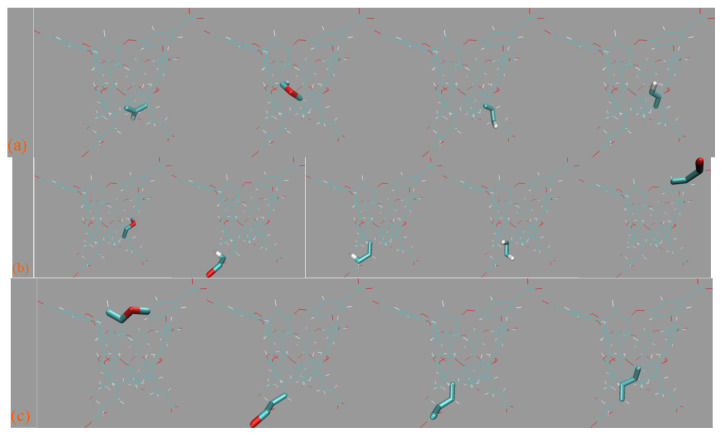
In the prototype OA, (**a**) torsional potentials that are exactly the same in the two GAFF versions, (**b**) those with altered barrier heights, and (**c**) terms newly defined in GAFF2. The first term in the subplot (**b**) has an elevated barrier height (almost ~2-fold increase) when shifting from GAFF to GAFF2, while the other four terms in GAFF2 have marginally altered barrier heights compared with GAFF. The differing terms located at the entrance of the OA cavity include the last term in the subplot b) and the first term in the subplot (**c**). The former in GAFF2 has a decreased barrier height (from 1 kcal/mol to 0.52 kcal/mol), producing slightly higher flexibility in the -COO^−^ region, and the latter with the ~0.9 kcal/mol barrier height stiffens the cavity entrance. The other newly added torsional potentials in GAFF2 strengthen the bottom of the host and make the dynamics there more regular.

**Figure 10 molecules-28-05940-f010:**
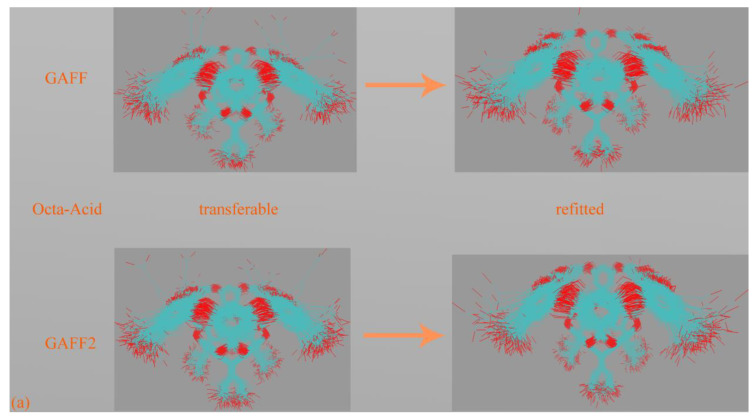
(**a**) Superpositions (5 ns per snapshot) of the host structure during the 500 ns explicit-solvent sampling of the solvated host under transferable and refitted force fields for the OA host. (**b**) Time series of the radius of gyration (Rg) of the host molecules (sampling interval 50 ps) with representative snapshots extracted from GAFF and GAFF2 trajectories. (**c**,**d**) Flipping of the benzoic acid groups with and without host–ion coordination.

## Data Availability

All parameter files of macrocyclic hosts before and after refitting are provided online at https://github.com/proszxppp/host-dynamics (accessed on 1 July 2023) for interested readers to probe the details of the force field differences and parameter variations upon regularized refitting and apply these high-quality parameter sets in their own research.

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
