# Peer review of "Host Dynamics under General-Purpose Force Fields"

_molecules, 2023, doi:10.3390/molecules28165940_

Round 1

Reviewer 1 Report

General comments for this work:

The study provides a comprehensive analysis of the energetics and dynamics of popular macrocycles, making significant contributions to the understanding of host-guest modelling in pharmaceutical research. The comparison of force-field definitions and parameters offers valuable insights into the potential impacts on the dynamics of macrocycles, aiding in force-field selection and parametrization. The utilization of the generalized force-matching scheme enhances the accuracy of force fields, increasing the reliability of the models. The findings shed light on the distinct behaviors exhibited by different macrocycles under general-purpose force fields, facilitating their characterization in terms of dynamics. The newly developed refitted force fields exhibit improved energetics and atomic forces when compared to the original GAFF and GAFF2, demonstrating the potential for enhanced modelling accuracy. The study's conclusions and recommendations, such as the preference for GAFF2 in practical applications, provide useful guidance for future research in host-guest modelling. Further investigations could explore the implications of these findings for other host families and address potential areas for improvement in force-field parametrization and modelling techniques. However, the manuscript could benefit from additional context and suggestions, as outlined below:

1.      In the introduction, it would be helpful to describe the specific methods used to investigate the energetics and dynamics of popular macrocycles in this study. Additionally, highlighting how these methods provide unprecedented detail would enhance the reader's understanding.

2.      In the methodology section, it would be beneficial to compare the AM1-BCC and RESP methods in terms of accuracy and computational efficiency for atomistic fixed-charge modelling. Providing references to support these comparisons would further assist the readers.

3.      Clarifying the differences and similarities between the two versions of GAFF (GAFF and GAFF2) in terms of bonded and van der Waals terms, and explaining how these differences potentially affect the dynamics of macrocycles, would improve the clarity of the paper and aid the audience's comprehension.

4.      Providing a description of how the generalized force-matching (FM) scheme works and its purpose in improving the accuracy of force fields would be beneficial. Including a scheme or workflow diagram (e.g., SCHEME 1 Diagram) could help visualize the FM scheme's process.

5.      In the discussion section, it would be valuable to elaborate on the observed dynamics of cucurbiturils, pillararenes, cyclodextrins, and octa acids under the general-purpose force fields. Identifying any significant differences or similarities in their behavior would enhance the understanding of these macrocycles' dynamics.

6.      Consider improving Figure 3 to enhance its clarity and effectiveness in conveying the intended information.

7.      It would be useful to compare the refitted force fields with the original GAFF and GAFF2 force fields in terms of energetics and atomic forces. This comparison would provide a clear understanding of the improvements achieved through the refitting process.

8.      Investigating the significance of differences in force-field definitions and parameters on the dynamics of macrocycles and providing general guidelines or recommendations for force-field parametrization and selection in host-guest modelling would be valuable. Supporting these guidelines with reasons and references would enhance the discussion.

9.      Specifically addressing the dynamics and energetics of sulfur-substituted pillararenes with -SO32- tails and comparing them to unmodified pillararenes would provide important insights for the readers.

10.  In the discussion section, it would be helpful to explain why GAFF2 is recommended as the preferred option for practical applications compared to GAFF, considering accuracy and overall performance. Including this explanation would provide a clear rationale for future researchers and readers.

Author Response

Reviewer #1:

General comments for this work:

The study provides a comprehensive analysis of the energetics and dynamics of popular macrocycles, making significant contributions to the understanding of host-guest modelling in pharmaceutical research. The comparison of force-field definitions and parameters offers valuable insights into the potential impacts on the dynamics of macrocycles, aiding in force-field selection and parametrization. The utilization of the generalized force-matching scheme enhances the accuracy of force fields, increasing the reliability of the models. The findings shed light on the distinct behaviors exhibited by different macrocycles under general-purpose force fields, facilitating their characterization in terms of dynamics. The newly developed refitted force fields exhibit improved energetics and atomic forces when compared to the original GAFF and GAFF2, demonstrating the potential for enhanced modelling accuracy. The study's conclusions and recommendations, such as the preference for GAFF2 in practical applications, provide useful guidance for future research in host-guest modelling. Further investigations could explore the implications of these findings for other host families and address potential areas for improvement in force-field parametrization and modelling techniques. However, the manuscript could benefit from additional context and suggestions, as outlined below:

Response: We are grateful to the kind comments provided to improve the quality of our research and valuable time spent by reviewers.

  1. In the introduction, it would be helpful to describe the specific methods used to investigate the energetics and dynamics of popular macrocycles in this study. Additionally, highlighting how these methods provide unprecedented detail would enhance the reader's understanding.

Response: The evaluation of energetic and dynamic behaviors is performed via unbiased sampling and QM calculations. As these are routine details that have been extensively discussed in following sections, we do not include them in the introduction part of the manuscript. 

  1. In the methodology section, it would be beneficial to compare the AM1-BCC and RESP methods in terms of accuracy and computational efficiency for atomistic fixed-charge modelling. Providing references to support these comparisons would further assist the readers.

Response: We already compare the accuracy of the two charge scheme in Fig. 2 (ESP RRMSE). As for the computational costs, AM1-BCC is obviously faster than RESP, as the former only involves gas-phase AM1 optimization, while the latter needs DFT or ab initio calculations for optimization and ESP scan. The BCC-faster-than-RESP fact has already been pointed out in the first paragraph of section 2 in the original submission.

  1. Clarifying the differences and similarities between the two versions of GAFF (GAFF and GAFF2) in terms of bonded and van der Waals terms, and explaining how these differences potentially affect the dynamics of macrocycles, would improve the clarity of the paper and aid the audience's comprehension.

Response: For CBn, the differing torsional terms have already been shown in Fig. 3a-b and discussed in section 3.1. For SPn, a detailed list/presentation of bonded parameters has been provided in Fig. 5, including a discussion about the variation of bond-stretching parameters in refitting (the mark in Fig. 5c). Similar analyses for methylated cyclodextrin and OA have been provided in Fig. 7 and Fig. 9. Therefore, the current version already provides detailed comparisons detailing the similarities and differences between GAFF and GAFF2.  

  1. Providing a description of how the generalized force-matching (FM) scheme works and its purpose in improving the accuracy of force fields would be beneficial. Including a scheme or workflow diagram (e.g., SCHEME 1 Diagram) could help visualize the FM scheme's process.

Response: The method works by performing a least-squares optimization of a given loss function including system energy, atomic force and regularization terms. Atom-type symmetry is involved intrinsically in the force-field definition. Adaptive optimization works by gradually adding new configurations to the training set. The workflow is not difficult to understand and the current manuscript already provides sufficient details. For readers interested in more detailed discussions and applications, we would recommend to read the original references of the method, i.e., ref. 37-38.

  1. In the discussion section, it would be valuable to elaborate on the observed dynamics of cucurbiturils, pillararenes, cyclodextrins, and octa acids under the general-purpose force fields. Identifying any significant differences or similarities in their behavior would enhance the understanding of these macrocycles' dynamics.

Response: Detailed discussions about host dynamics of cucurbiturils, pillararenes, cyclodextrins and octa acids have been provided in section 3.1-3.4 and illustrative pictures of GAFF(2) dynamics and refitting-induced variations are presented in Fig. 3-10 and the supplementary material. In the revised manuscript, we include additional results for the CB10 system (Fig. 4).

  1. Consider improving Figure 3 to enhance its clarity and effectiveness in conveying the intended information.

Response: We increased the color contrast by brightening the texts and darkening the background color of Fig. 3 in revision.

  1. It would be useful to compare the refitted force fields with the original GAFF and GAFF2 force fields in terms of energetics and atomic forces. This comparison would provide a clear understanding of the improvements achieved through the refitting process.

Response: We already present such comparisons in Fig. S1, S3, S5, and S6.

  1. Investigating the significance of differences in force-field definitions and parameters on the dynamics of macrocycles and providing general guidelines or recommendations for force-field parametrization and selection in host-guest modelling would be valuable. Supporting these guidelines with reasons and references would enhance the discussion.

Response: The ultimate reference in this work is the dynamic behavior produced by the high-accuracy refitted FM-B97-3c force fields. These refitted parameters are specific to individual molecules and thus have more degrees of freedom compared with transferable force fields. The closeness of the GAFF/GAFF2 dynamics to the FM-B97-3c results is the measure of the quality of the force field. In the investigation of all host molecules, we are providing the recommendations based on this protocol and already discuss this in great details.

  1. Specifically addressing the dynamics and energetics of sulfur-substituted pillararenes with -SO32- tails and comparing them to unmodified pillararenes would provide important insights for the readers.

Response: Unmodified pillararenes are not actually of great pharmaceutical interest due to their low solubility in water. As a result, chemically modified species such as the -SO32- and -CH2COO- groups are commonly introduced to improve the solubility and ultimately make the host-guest complex soluble in water. Therefore, the prototype is of relatively small importance in practical pharmaceutical researches and that’s the reason for selecting sulfur-substituted pillararenes as the investigation target of the current research. Further, the difference between the prototype and the -SO32- substituted form differs only in the substituting sites and thus the difference between host dynamics is rather clear. I.e., the behavior of the prototype can be derived from the backbone dynamics of SPn. 

  1. In the discussion section, it would be helpful to explain why GAFF2 is recommended as the preferred option for practical applications compared to GAFF, considering accuracy and overall performance. Including this explanation would provide a clear rationale for future researchers and readers.

Response: We already discuss the criteria of selecting force fields in great details in the existing version. Specifically, using the energetic and dynamic data of FM-B97-3c parameter sets as the reference, we recommend using transferable force fields that reproduce the reference results. Based on this protocol, for cucurbiturils, the GAFF2 parameter set is preferred for large rings such as CB8, but both GAFF and GAFF2 are usable for the small CB7, as the force-field differences do not lead to noticeable alterations of host dynamics. Further, it should be noted that for further enlarged rings such as CB10, GAFF2 also seems slightly soft and produce host dynamics obviously deviating from the FM-B97-3c reference (see Fig. 4). Thus, neither transferable force fields are sufficiently accurate. For sulfur-substituted pillararenes, the host dynamics under GAFF and GAFF2 are extremely similar, and using either of them in practical applications is fine. Thus, there is no GAFF2-better-than-GAFF behavior for SPn. For methylated cyclodextrins, neither GAFF nor GAFF2 is good enough, and using refitted force fields is necessary for accurate host dynamics. For the octa acid, the GAFF and GAFF2 results are also similar and thus both of them are usable. The details of the comparison of force-field definitions, energetic and dynamic perspectives are already provided in section 3.

Reviewer 2 Report

See file attached.

Author Response

Reviewer #2:

Review: molecules-2526586

This paper presents some potentially interesting results for macrocyclic host-guest compounds using general purpose forcefields. In principle the work is compelling for chemists, chemical engineers, and chemical physicists and it can inform future force field development as well as future research on macrocycles. The authors present structural, configurational, and energetic results in their discussion and analysis of various popular macrocycles.

I would recommend publication of this paper, although as far as I can tell, key elements of the methodology are not described in sufficient detail:

Response: The manuscript has been revised following reviewer’s comments. We further added new experiments on the 10-unit CBn specie in section 3, revealing really interesting behaviors and identifying weaknesses of both GAFF and GAFF2 (existing transferable force fields).

  1. Firstly, did the authors perform multiple independent simulations or are the results each from single simulation runs? Multiple independent simulations with uncertainty estimates, especially for configurational data such as radius of gyration (and possible inclusion of statistical comparison testing) may reveal statistically significant or insignificant differences that are difficult to elucidate from the graphs presented.

Response: We only perform a single 500 ns run for each parameter set. The simulation tool is approximating the ensemble average with the time-series data, and this is not influenced by the number of replaces. As long as a single simulation is sufficiently long, the converged statistics serve as the reliable estimates of the system properties. The reasoning of using multiple trajectories is considering them as independent experiments, in a spirit similar to experimental measurements. Our 500 ns sampling length is already sufficiently long to probe host dynamics and there is no need to perform multiple runs.

  1. Secondly, what thermostats and barostats were employed in this study? The authors must present more computational details in order for others to replicate, and ultimately expand on the work shown.

Perhaps I am missing something, but I did not see clear mention of the temperature and/or pressure control schemes used in the simulations, nor details on their associated parameters.

Response: Brute-force simulations are routine and easy to set up, and thus we did not include the details in the first version. In the revised manuscript, we provide these details in the CBn section (i.e., section 3). I put the basic details below for easier access: GROMACS 2020.6, velocity rescaling for temperature regulation at 300 K, Parrinello-Rahman barostat for pressure regulation at 1 atm, a time step of 1 fs, smooth Particle-mesh Ewald for long-range electrostatics.

In addition the above comments, there are some minor points raised below:

  1. Page 3, line 81: It is redundant to say “fixed-charge modelling models”, you can just say fixed charge models”.

Response: The original sentence is correct. ‘models’ is a verb here.

  1. Pages 7-8, Figure 3: The a), b), and c) are not easy to read as they are blue text on a grey background; perhaps consider white text or perhaps make the background white. In addition, the text in 3c) exhibits the same problems with being easy to read. Consider revising the colour and aim for higher colour contrast.

Response: We increased the color contrast by brightening the texts and darkening the background color.

  1. Page 12, line 392: It looks like a word is missing; should it not be “…applied in many pharmaceutical research studies” or something similar?

Response: The sentence has been modified in the revised manuscript.

  1. Page 14, Figure 5 a): The text is difficult to read; try to increase the colour contrast.

Response: We increased the color contrast by darkening the background color of Fig. 5a.

  1. Page 18, Figure 7 a): The text is difficult to read; try to increase the colour contrast

Response: We increased the color contrast by darkening the background color.

Reviewer 3 Report

The authors presented a detailed study on examining existing and parametrizing new force field description on structural dynamics of several example host molecules. The authors confirmed the accuracy of the existing force fields for the overall dynamics of the molecules while showed discrepancies on the bonding interactions. This is in consistent with the usually observed point-wise force discrepancies due to subtle geometry discrepancies between ab initio and force-field levels of theory. Generally I found the paper comprehensive and encouraged to publish in the current form. Publish as it is.

Minor changes needed.

Author Response

Reviewer #3:

The authors presented a detailed study on examining existing and parametrizing new force field description on structural dynamics of several example host molecules. The authors confirmed the accuracy of the existing force fields for the overall dynamics of the molecules while showed discrepancies on the bonding interactions. This is in consistent with the usually observed point-wise force discrepancies due to subtle geometry discrepancies between ab initio and force-field levels of theory. Generally I found the paper comprehensive and encouraged to publish in the current form. Publish as it is.

Response: We are grateful to the kind comments provided to improve the quality of our research and valuable time spent by reviewers.

Round 2

Reviewer 1 Report

The study is revised well and can be accepted. 

Reviewer 2 Report

The authors seem to have addressed the concerns raised.